

# Comparative study of the gut microbial community structure of *Spodoptera frugiperda* and *Spodoptera literal* (Lepidoptera)

Yaping Chen[1,*], Yao Chen[1,*], Yahong Li[2], Ewei Du[1], Zhongxiang Sun[1], Zhihui Lu[1] and Furong Gui[1]

[1] College of Plant Protection, Yunnan Agricultural University, Kunming, Yunnan, China
[2] Yunnan Plant Protection and Quarantine Station, Kunming, Yunnan, China
[*] These authors contributed equally to this work.

Corresponding author
Furong Gui, guifr@ynau.edu.cn

## ABSTRACT

**Background**. *Spodoptera frugiperda*, the fall armyworm is a destructive invasive pest, and *S. litura* the tobacco cutworm, is a native species closely related to *S. frugiperda*. The gut microbiota plays a vital role in insect growth, development, metabolism and immune system. Research on the competition between invasive species and closely related native species has focused on differences in the adaptability of insects to the environment. Little is known about gut symbiotic microbe composition and its role in influencing competitive differences between these two insects.

**Methods**. We used a culture-independent approach targeting the 16S rRNA gene of gut bacteria of 5th instar larvae of *S. frugiperda* and *S. litura*. Larvae were reared continuously on maize leaves for five generations. We analyzed the composition, abundance, diversity, and metabolic function of gut microbiomes of *S. frugiperda* and *S. litura* larvae.

**Results**. Firmicutes, Proteobacteria, and Bacteroidetes were the dominant bacterial phyla in both species. *Enterococcus*, *ZOR0006*, *Escherichia*, *Bacteroides*, and *Lactobacillus* were the genera with the highest abundance in *S. frugiperda*. *Enterococcus*, *Erysipelatoclostridium*, *ZOR0006*, *Enterobacter*, and *Bacteroides* had the highest abundance in *S. litura*. According to α-diversity analysis, the gut bacterial diversity of *S. frugiperda* was significantly higher than that of *S. litura*. KEGG analysis showed 15 significant differences in metabolic pathways between *S. frugiperda* and *S. litura* gut bacteria, including transcription, cell growth and death, excretory system and circulatory system pathways.

**Conclusion**. In the same habitat, the larvae of *S. frugiperda* and *S. litura* showed significant differences in gut bacterial diversity and community composition. Regarding the composition and function of gut bacteria, the invasive species *S. frugiperda* may have a competitive advantage over *S. litura*. This study provides a foundation for developing control strategies for *S. frugiperda* and *S. litura*.

## INTRODUCTION

The fall armyworm, *Spodoptera frugiperda* (J.E. Smith, 1797), is a destructive invasive pest originating from tropical and subtropical America (*Todde & Poole, 1980*). In 2016, *S. frugiperda* invaded Africa from America and spread to more than 100 countries, including India, Bangladesh, Sri Lanka, Thailand, and Myanmar (*CABI, 2019*). In January 2019, *S. frugiperda* was first reported in Jiangcheng, Pu'er City, Yunnan Province in China (*Jing et al., 2020*). Since then, *S. frugiperda* has spread rapidly to 26 provinces (*Wang, Chen & Lu, 2019*). *S. frugiperda* larvae attack more than 350 host plant species belonging to 76 plant families, including maize, rice, sugarcane, sorghum, peanut, buckwheat, lettuce, and cotton, but the greatest damage is observed in maize (*Montezano et al., 2018*). In China, the larvae of *S. frugiperda* are causing 20–30% crop loss in maize, leading to extensive economic losses and threatening food security (*Mao, 2019*).

*S. litura* (Fabricius, 1775) is native to China and closely related to *S. frugiperda*. After *S. frugiperda* invaded China, it usually co-occurs with *S. litura* in maize fields (*Zhao, Luo & Sun, 2019*). Because they share the host species, the two pests' frequency of occurrence, and morphological characteristics are very similar. Therefore, it is challenging to formulate different control strategies for the native and the invasive species. In an earlier study, we found that cytochrome P450 (CYP) genes are more expanded in *S. frugiperda* than in *S. litura* and six other Lepidopteran species, and the expanded CYP genes from *S. frugiperda* showed very short time divergence (*Gui et al., 2020*). Moreover, compared to *S. litura*, GSTs in certain branches showed obvious expansion, identifying several amino acid mutations, some of which were predicted to affect protein function (*Huang et al., 2019*; *Zhu et al., 2020*). These characteristics help *S. frugiperda* deal with a wider variety of exogenous toxic and odorous substances, which may be one of the reasons for its rapid invasion, giving it a competitive advantage over native species.

The gut microbiome is a complex ecosystem dominated by bacteria, forming a close symbiotic relationship with the host (*Clemente et al., 2012*). Insect gut bacterial community structure differs greatly between insect species and even among individuals. Gut bacteria can help insects digest food, improve protection against pathogens and pesticides, regulate mating and reproductive functions, degrade toxic substances in host plants, and affect intraspecific and interspecific substance exchange (*Zhou et al., 2020*; *Chen, Lu & Shao, 2017*; *Zilber-Rosenberg & Rosenberg, 2008*). For instance, the bessbug's (*Odontotaenius disjunctus*; a wood-feeding insect) gut structure and microbial composition contribute to the decomposition of lignocellulose, which helps *O. disjunctus* digest food (*Ceja-Navarro et al., 2019*). In the gut of *S. frugiperda* larvae, a variety of bacteria such as *Klebsiella*, *Staphylococcus*, *Bacillus*, and *Acinetobacter* can produce cellulase, xylanase, pectinase, and metabolize phenol, improving its adaptability to the host plants (*Chen et al., 2022*). Furthermore, a causal link exists between gut microbial composition and insect resistance to insecticides. Gut bacteria may prevent or restore damage to insect immune systems caused by insecticides rather than bacterial degradation, constituting the mechanism of acquired resistance (*Shao et al., 2014*).

Studies on the competition between invasive species and closely related native species have focused on the differences in the adaptability of insects to the environment, while little is known about the differences in and roles of gut symbiotic microbes (*Huang et al., 2019*; *Zhu et al., 2020*). Food is an important factor in the interspecific competition (*Farjana, Tuno & Higa, 2012*), and food is also one of the most important factors affecting the gut bacterial community structure (*Yun et al., 2014*). Therefore, we wanted to explore differences in the gut bacterial structure between *S. frugiperda* and *S. litura* under the influence of the same host plant, and the potential effects these differences may have on their competitive relationship.

Here, we used 16S rRNA sequence profiling to characterize the diversity of gut microbiota associated with *S. frugiperda* and *S. litura* that fed on maize leaves, reporting gut bacterial species. Then, we preliminarily explored the commonalities and differences between the gut bacteria communities of these two insect species. Functional bacterial species especially play an essential role in the host fitness (*Armitage et al., 2022*). Our findings provide a basis for future studies to develop novel pest management strategies, especially by formulating specific gut microecological regulation strategies.

## MATERIALS & METHODS

### Source of insects for testing

Larvae of *S. frugiperda* and *S. litura* were collected from a corn field in Yuanjiang County, Yuxi City, Yunnan Province (101°58′E, 25°35′ N, 421 m a.s.l.) in May 2019. Both species were raised in artificial climate chambers (MG-300A, Shanghai Yiheng Scientific Instrument Co., Ltd., Shanghai, China) at a temperature of $27 \pm 0.5$ °C, a 16:8 h light-dark cycle, and a relative humidity of $70 \pm 5\%$ (*Chen et al., 2022*). After the larvae pupated, the pupae were placed in a finger tube for independent observation. After emergence, the adults of both species were paired with a female-to-male ratio of 1:1. The paired males and females were placed in an egg-laying box. The box was filled with absorbent cotton containing 10% honey water to provide sufficient nutrition for the adults. After oviposition, the eggs of the same female were retrieved separately for testing. The larvae were raised on an artificial diet for 12 generations in a rearing box to eliminate the effect of pesticides in the wild on gut bacterial community structure. Maize seeds were sown in flowerpots without using any pesticides. The larvae of both species were fed fresh maize leaves for five generations.

### Sample collection

The 5th instar larvae were starved for 24 h before collection of the midgut. The larvae were surface-sterilized with 75% alcohol for 120 s, rinsed with sterilized-ultrapure water three times, and then dissected. The entire midgut tissues of three larvae were collected in a centrifuge tube filled with 5 ml sterile phosphate buffer saline (PBS), thoroughly shaken with an oscillator, and mixed to prepare a suspension of gut contents. Each treatment was set up with eight replicates, and each replicate collected the midgut tissues with the content of three larvae.

## DNA extraction and sequencing

Total DNA was extracted from the gut using a QIAamp DNA Stool Mini kit (Qiagen, Hilden, Germany). The ratio of A260/A280 for all samples ranged between 1.7 and 1.9. The 16S rRNA gene was sequenced in the V3-V4 region using an Illumina MiSeq/HiSeq sequencing platform.

## Sequencing data statistics and quality control

Raw sequence data were processed using the software Trimmomatic (version 0.35) (*Bolger, Lohse & Usadel, 2022*) to detect and cut ambiguous bases. After trimming low-quality sequences, paired-end reads were assembled using the software FLASH (version 1.2.11) based on a maximum overlap of 200 bp (*Reyon et al., 2014*). High-quality clean tags were obtained by the software split libraries (version 1.8.0) in QIIME (*Caporaso et al., 2010*) for subsequent analysis by removing the sequences containing N bases, sequences with single base repetitions greater than eight, and sequences with lengths greater than 200 bp.

## Identification and diversity analysis of gut bacteria

Using Usearch software, all sample reads of samples were clustered at a similarity level of 97.0% to obtain operational taxonomic units (OTUs), and the relative abundance of species at each taxonomic level was counted in each sample. The species diversity of the samples was evaluated by calculating species richness, Shannon, Chao1, and Simpson indices using the software QIIME. Similarities and differences between the samples were analyzed using principal coordinates analysis (PCoA). Based on the Kyoto Encyclopedia of Genes and Genomes (KEGG) database, functional prediction analysis of 16S rDNA sequencing data was performed using the PICRUSt tool (*Douglas et al., 2020*; *Langille et al., 2013*) and visualized using the software STAMP.

# RESULTS

## Sequence splicing assembly and OTU clustering analysis

The Illumina MiSeq/HiSeq sequencing platform was used to sequence the V3-V4 region of the 16S rDNA of maize-fed *S. frugiperda* and *S. litura*. The average number of optimized sequences obtained for *S. frugiperda* (65,031) was lower than that for *S. litura* (66,532), with an average of 79,678 and 79,619 pairs of reads obtained for *S. frugiperda* and *S. litura* samples, representing 97.8% and 97.9% of the valid sequences. The sequencing accuracy of the samples was satisfactory and met the analysis requirements. The number of OTUs of *S. frugiperda* samples was higher than that of the *S. litura* samples (Figs. 1A–1B).

## Analysis of the diversity of the *S. frugiperda* and *S. litura* samples

The difference in gut microbial $\alpha$-diversity between *S. frugiperda* and *S. litura* was analyzed using the Shannon, Simpson, ACE, and Chao 1 indices regarding species richness and evenness. The Shannon indices of *S. frugiperda* were significantly higher than those of *S. litura* ($p < 0.05$; Figs. 1C–1F). These results suggested that gut microbiota diversity was significantly higher in *S. frugiperda* than in *S. litura*.

Weighted-Unifrac distance between each sample was used for $\beta$-diversity analyses. In PCoA and NMDS, samples of *S. frugiperda* clustered separately from those of *S. litura*.

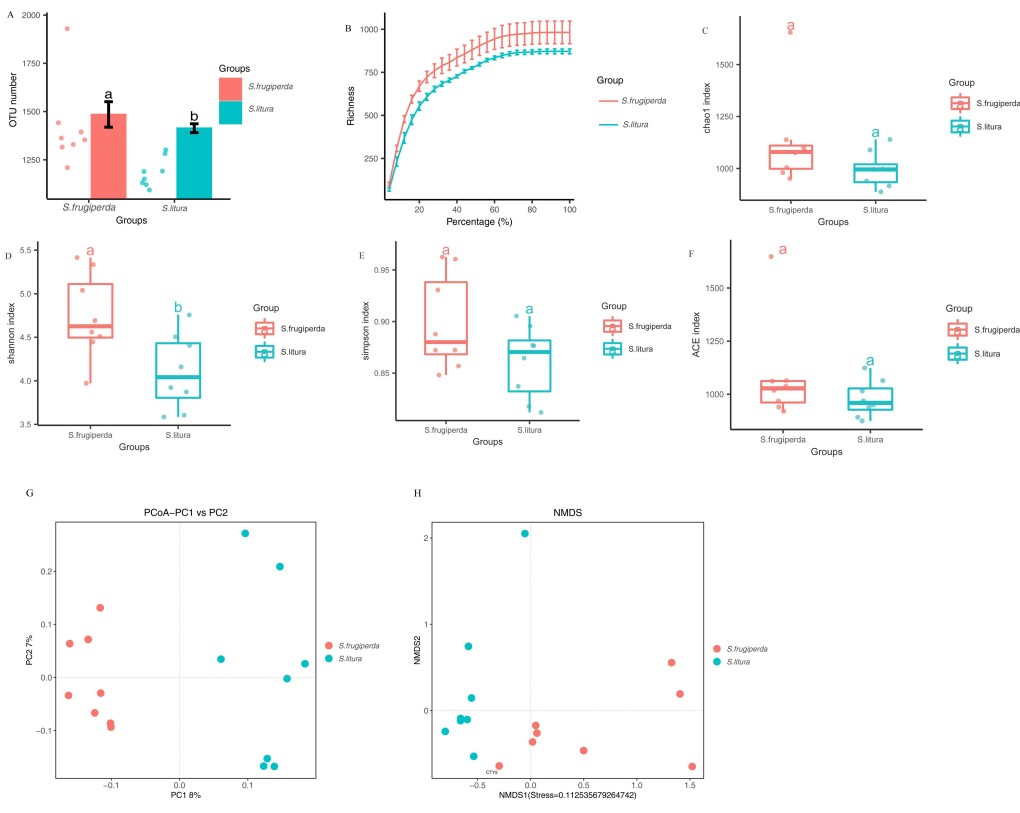

**Figure 1** **Comparison of gut microbial diversity and differences between *S. frugiperda* and *S. litura*.** (A) Columnar scatter plot of the number of OTUs. (B) Exponential dilution curve. (C–F) Bacterial Alpha diversity index of different samples. (G, H) Unifrac principal coordinate analysis (PCoA) and nonmetric multidimensional scaling (NMDS) of different samples.

Through $\beta$-diversity analyses using ANOSIM, we found that the overall structure of the gut microbiome in *S. frugiperda* and *S. litura* were significantly different (Figs. 1G–1H).

## Species difference analysis

The three phyla with the highest abundances in *S. frugiperda* and *S. litura* samples were Firmicutes (50.7 and 55.6%, respectively), Proteobacteria (22.4 and 16.9%) and Bacteroidetes (16.2 and 12.6%) (Figs. 2A–2B). The ten genera with the highest abundances were *Enterococcus*, *ZOR0006*, *Enterobacter*, *Erysipelatoclostridium*, *Bacteroides*, *Escherichia*, *Prevotella.9*, *Lactobacillus*, *Ruminoccaceae.UCG.014*, and *Sphingomonas*. Phylogenetic analysis was conducted based on the top 10 OTUs in all samples at the genus level (Fig. S1). In *S. frugiperda*, *Enterococcus* (27.4%), *ZOR0006* (9.4%), *Escherichia* (3.5%), *Bacteroides* (2.2%), and *Lactobacillus* (1.5%) were the five species with the highest abundances. In *S. litura*, *Enterococcus* (34.5%), *Erysipelatoclostridium* (8.8%), *ZOR0006* (1.9%), *Enterobacter* (1.8%), and *Bacteroides* (1.7%) had the highest abundances (Figs. 2C–2D).

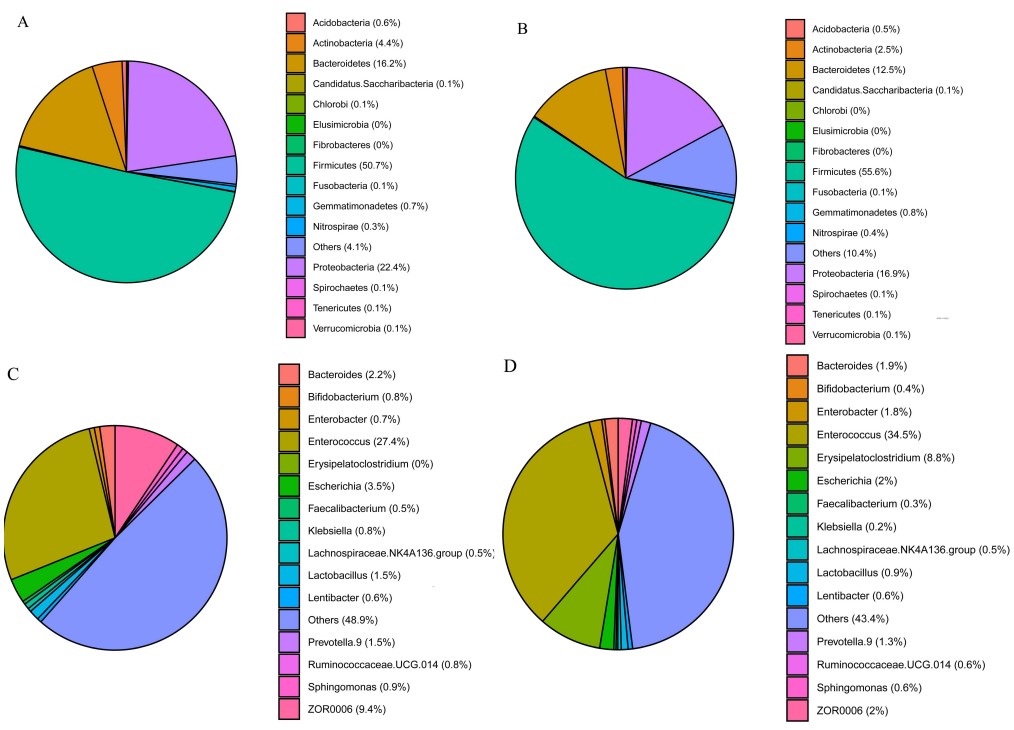

**Figure 2 Gut bacterial composition of *S. frugiperda* and *S. litura*.** (A, B) Relative abundance at phylum level in *S. frugiperda* (A) and *S. litura* (B). (C, D) Relative abundance at genus level in *S. frugiperda* (C) and *S. litura* (D).

## Analysis of *S. frugiperda* and *S. litura* larval gut microbiome biomarkers

To further analyze the differences in gut bacteria between samples and to find key microbiome elements, the abundance profiles of OTUs obtained after species annotation were analyzed for linear discriminant analysis effect values (LEfSe) to find species that differed significantly in abundance between pest species (*i.e.,* biomarkers). The number of biomarkers at different taxonomic levels that differed significantly between *S. frugiperda* and *S. litura* was 29 and 3 on genus and phylum level, respectively (Fig. 3A). There were more biomarkers in *S. frugiperda*. The biomarkers for *S. litura* included *Erysipelatoclostridium*, Firmicutes, and *Enterobacter*. The biomarkers for *S. frugiperda* were dominated by the phyla Actinobacteria, Flavobacteria, Proteobacteria, Bacteroidetes and Firmicutes, including Micrococcaceae, *Leucobacter*, *Bifidobacterium*, Flavobacteriaceae, *Pseudochrobactrum*, *Ochrobactrum*, Muribaculaceae, *Escherichia*, Brucellaceae, Bacteroidales, *ZOR0006*, *Lactobacillus*, Peptostreptococcaceae, and Clostridiales (Fig. 3A). Because of different taxonomic levels of biomarkers, the evolutionary relationships of these biomarkers were shown in Fig. 3B and Table S1.

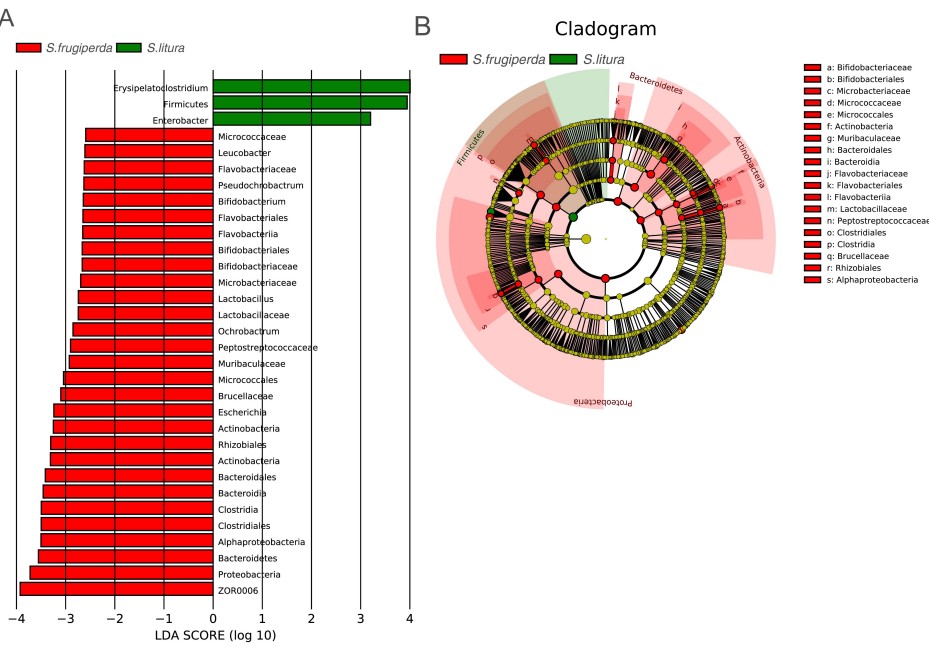

**Figure 3** Gut microbiome biomarkers of *S. frugiperda* (left) and *S. litura* (right). (A) All the bar charts show those taxa that were significantly differentially abundant between comparison groups. All the taxa are ranked by effect size, and only taxa meeting an LDA significance threshold of >2.0 are shown. (B) The taxonomic cladograms obtained from LEfSe analysis of 16S rDNA sequences are shown. Small circles highlighted in different colors (red and green) in the diagram represent the taxa that were significantly elevated in the respective group. Yellow circles indicate taxa that were not significantly differentially represented (*P* > 0.05).

## Functional predictions and differences in the gut microbiome of *S. frugiperda* and *S. litura*

We compared the COG function terms and KEGG pathways between *S. frugiperda* and *S. litura* and visualized the results using STAMP software. The two pest species had 15 significantly different KEGG function terms (Figs. 4A–4B). The dominant five functional categories were transcription, cell growth and death, excretory system, cardiovascular diseases, and circulatory system. COG pathway analysis showed 13 significantly different metabolic pathways between the two species. Further, PCA illustrated functional differences between the two groups (Fig. 4C).

## DISCUSSION

*S. litura* is the native species most closely related to *S. frugiperda*. The two species may compete in the same environment due to their similar ecological niche (*Montezano et al., 2018*; *Tang et al., 2022*). In species competition, one species may crowd out or even replace other species, and if invasive species crowd out native species, native species diversity is affected. In this study, the diversity and richness of the gut bacterial communities of *S. frugiperda* and *S. litura* were studied, functions of the gut bacterial community were predicted, and differences in the structure and function of the gut

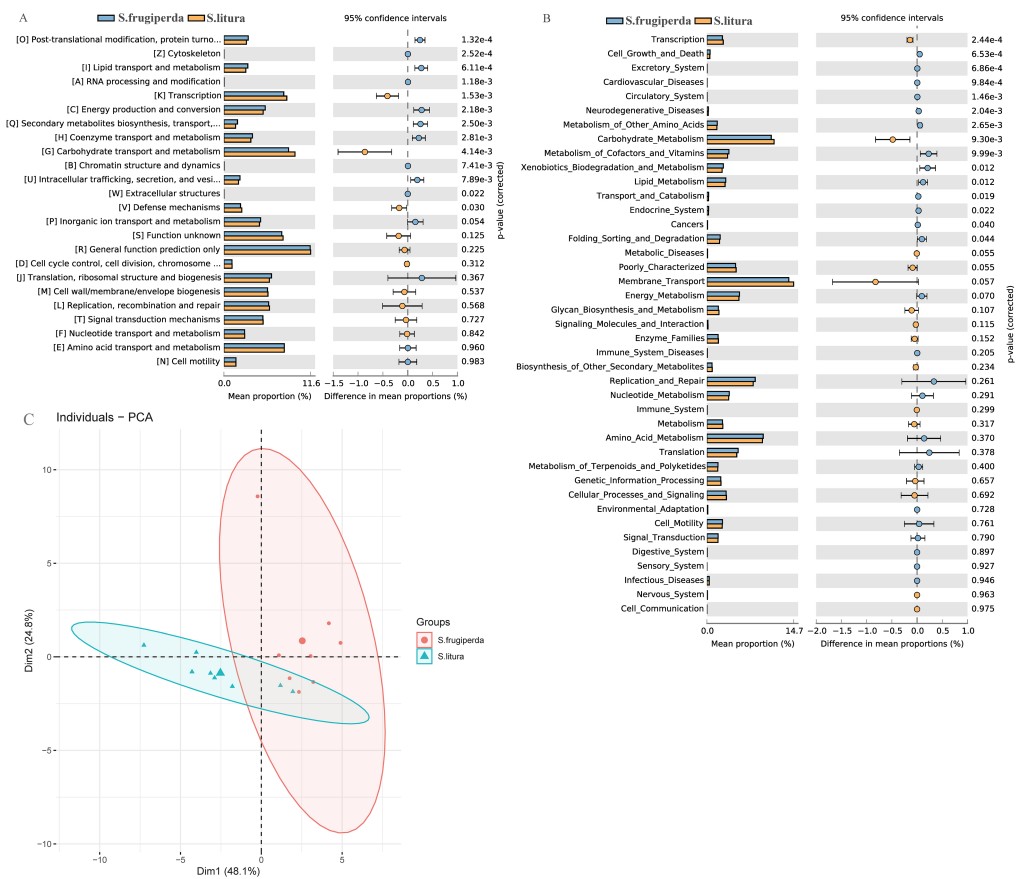

**Figure 4 Functional predictions and differences in the gut flora of *S. frugiperda* and *S. litura*.** (A) Clusters of Orthologous Group (COG) function classification. (B) Kyoto Encyclopedia of Genes and Genomes (KEGG) function classification. (C) Principal component analysis (PCA) of KEGG.

bacterial community between the two species were found. This is significant for analyzing differences in environmental adaptation between invasive species and their native relatives and, ultimately, managing invasive species.

We found that the gut bacterial communities of *S. frugiperda* and *S. litura* were dominated by Firmicutes, Proteobacteria, and Bacteroidetes at the phylum levels (Figs. 2A–2B). This aligns with findings on other Lepidoptera, Hymenoptera, and Coleoptera *e.g.*, *Bombyx mori*, *Lymantria dispar*, *Helicoverpa armigera*, Apis, and *Holotrichia oblita* and may be the result of long-term insect evolution (*Sheng et al., 2012*; *Mason et al., 2016*; *Kumar et al., 2019*; *Frago, Dicke & Godfray, 2012*; *Martinson, Moy & Moran, 2012*). Herbivorous insects need carbohydrate hydrolases to help them digest and degrade host plants (*Ceja-Navarro et al., 2019*), and the enzymes that degrade cellulose, hemicellulose, and pectin are mainly encoded by the bacterial genes of Firmicutes and Proteobacteria (*Dantur et al., 2015*). They play a vital role in maintaining their normal physiological functions.

At the genus level, *Enterococcus* had the highest abundance of gut bacteria in both insect species (Figs. 2C–2D). *Enterococcus* is abundant in many insect guts and plays an important role in detoxifying pesticides and providing a natural defense system (*Shao et al., 2014*; *Vilanova et al., 2016*). *Enterococcus* contributed 82% to gut bacterial abundance in silkworms, effectively preventing the colonization of pathogenic bacteria and maintaining the balance of the gut microbiome (*Zhang et al., 2022*). *Enterococcus* isolated from the gut of *Brithys crini* can effectively degrade plant terpenes and reduce their toxic effect on insects (*Vilanova et al., 2016*). *Enterococcus* from *Plutella xylostella* can significantly enhance this moth's resistance to chlorpyrifos insecticides (*Hadi et al., 2021*). In the gut of *S. litura*, *Enterococcus* is metabolically active and can consume plant cellulose quickly, releasing glucose as an energy source for both the host insect and itself (*Devi et al., 2022*). The abundance of *Enterococcus* was significantly higher in the gut of *S. litura* than in that of *S. frugiperda*. *Enterococcus* can form a biofilm on the gut epidermis of *S. litura* larvae, acting as a shield against foreign potentially harmful microorganisms (*Shao et al., 2014*).

There were significantly fewer biomarkers in *S. litura* than in *S. frugiperda*, and *Enterobacteria* and *Erysipelatoclostridium* were common gut strains of *S. litura*. Biomarkers for *S. frugiperda* included Micrococcaceae, *Leucobacter*, and *Bifidobacterium* belong to the Actinobacteria (Fig. 3A), which are prevalent in the guts of various animals and insects and can help the host degrade toxic substances produced by oxygen-mediated respiration (*Bottacini et al., 2012*). Flavobacteriaceae, a common symbiote of insects, can help insects synthesize essential amino acids and complete phylogeny (*Short et al., 2017*). Muribaculaceae, Brucellaceae, *Ochrobactrum*, *Pseudochrobactrum*, and *Escherichia* belong to the Proteobacteria. *Ochrobactrum* can degrade pesticides and help Termitidae degrade lignin, cellulose, and hemicellulose (*Fathollahi et al., 2021*). Brucellaceae may be related to the metabolism of heavy metals in insects (*Wu et al., 2020*). Bacteroidales of Bacteroidetes can help insects produce specific enzymes for nitrogen metabolism under a nitrogen-deficient diet (*Desai & Brune, 2012*). Peptostreptococcaceae, Clostridiales, *ZOR0006,* and *Lactobacillus* belong to the Firmicutes. During the rainy season, *ZOR0006* increased in relative abundance in *S. frugiperda*. This may be related to *S. frugiperda*'s temperature regulation (*Higuita Palacio et al., 2021*). *Lactobacillus* is one of the most common probiotics in humans and one of the main symbiotic bacteria of insects. It assists in the physiological development of *Drosophila* larvae (Table S1) (*Storelli et al., 2011*).

Through functional analysis using the KEGG, it was found that *S. frugiperda* had the higher abudance of genes that participated in most metabolic pathways than that of *S. litura*. The gut bacteria of *S. frugiperda* were enriched in genes with nutrient metabolism function. These genes mainly exercise metabolic functions regarding amino acid, vitamin, and lipid metabolism (Figs. 4A–4B). Amino acids and other nutrients have low concentrations in plants and often cannot fully meet insects' growth and development requirements. However, insect gut bacteria can synthesize some of these nutrients. This phenomenon has been confirmed in studies of *Plutella xylostella* (*Indiragandhi et al., 2007*). COG enrichment analysis showed similar results, with significant differences between *S. frugiperda* and *S. litura* in 13 COG functional classifications. The most significant differences occurred in post-translational modification, cytoskeleton, lipid transport and metabolism, and

RNA processing. The relatively high abundance of modifications indicated that the gut bacteria of *S. frugiperda* have strong anabolism and transportation functions (Fig. 4A). This phenomenon may be due to the long evolution and adaptation of gut bacteria in insects, as well as the development of strategies to cope with environmental stress. Based on $\beta$ diversity analysis, the survival requirements of the two species led to a differentiation of gut bacterial communities and functions (Fig. 4C). *S. frugiperda* had significantly more functional genes than *S. litura*, which may give *S. frugiperda* a greater potential for adaptation to host plants and a competitive advantage.

## CONCLUSIONS

In conclusion, our systematic comparison between *S. frugiperda* and *S. litura* gut microbiomes showed that distinctive *S. frugiperda* and *S. litura* gut bacterial strains possess different functions, possibly related to their inter-species diversity. This study laid the foundation for further research on the function of gut bacteria in *S. frugiperda* and *S. litura* and adaptation mechanisms to host plants. The comparison between invasive and native species is conducive to differential pest control, and *S. frugiperda* may have a competitive advantage over *S. litura* regarding the functional characteristics of the gut microbiome.

### Funding

This study was funded by the National Key R&D Program of China (grant No. 2021YFD1400701), and the National Natural Science Foundation of China (grant No. 32360668). The funders had no role in study design, data collection and analysis, decision to publish, or preparation of the manuscript.

### Grant Disclosures

The following grant information was disclosed by the authors:
National Key R&D Program of China: 2021YFD1400701.
National Natural Science Foundation of China: 32360668.

### Competing Interests

The authors declare there are no competing interests.

### Author Contributions

- Yaping Chen conceived and designed the experiments, performed the experiments, analyzed the data, prepared figures and/or tables, authored or reviewed drafts of the article, and approved the final draft.
- Yao Chen conceived and designed the experiments, performed the experiments, analyzed the data, prepared figures and/or tables, authored or reviewed drafts of the article, and approved the final draft.
- Yahong Li performed the experiments, prepared figures and/or tables, and approved the final draft.
- Ewei Du conceived and designed the experiments, analyzed the data, prepared figures and/or tables, and approved the final draft.
- Zhongxiang Sun analyzed the data, authored or reviewed drafts of the article, and approved the final draft.
- Zhihui Lu performed the experiments, analyzed the data, prepared figures and/or tables, authored or reviewed drafts of the article, and approved the final draft.
- Furong Gui conceived and designed the experiments, authored or reviewed drafts of the article, and approved the final draft.

## Data Availability

The original data for the tobacco cutworm and the fall armyworm are available at NCBI SRA: PRJNA107608.

https://www.ncbi.nlm.nih.gov/bioproject/PRJNA1076080/.

## Supplemental Information

Supplemental information for this article can be found online at http://dx.doi.org/10.7717/peerj.17450#supplemental-information.

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
