# Peer review of "Comparative study of the gut microbial community structure of Spodoptera frugiperda and Spodoptera literal (Lepidoptera)"

_PeerJ, doi:10.7717/peerj.17450_

## Round 0.1 · original submission · Major Revisions

Two experts assessed your manuscript and found the content relevant. Some concerns, though, need to be addressed before reaching a final call. Among them, confirming the findings on gut microbiome using wild animals is a key point to be addressed. Please also note the advice on improving English usage.

**Language Note:** The Academic Editor has identified that the English language must be improved. PeerJ can provide language editing services - please contact us at copyediting@peerj.com for pricing (be sure to provide your manuscript number and title). Alternatively, you should make your own arrangements to improve the language quality and provide details in your response letter. – PeerJ Staff

Reviewer 1 ·

Basic reporting

This manuscript entitled “Comparative study on the gut microbial community structure of Spodoptera frugiperda and S. litura (Lepidoptera)“ reports the role of gut microbiota in the invasive success of agricultural pests. The authors sequenced the 16S rRNA of gut bacteria of 5th instar larvae of S. frugiperda and S. litura to report differences in the gut microbial diversity and composition between the two insect species. Based on this data, the authors suggest that gut-microbiota may provide a competitive advantage to S. frugiperda over S. litura. The manuscript demands thorough revision. The language demands editing. The authors should pay attention to italicizing scientific names throughout the manuscript. However, the references, figures, tables are satisfactory.

Experimental design

The research question is well defined, relevant & meaningful. The following points may be considered:

1. Larvae of S. frugiperda and S. litura were collected from corn field and reared in lab conditions for multiple generations before gut-microbial analysis. The authors have however not justified selection of this particular experimental strategy for sequencing.

2. It is not clear if the differences in the gut microbiota of the two Spodoptera species also exist in the natural conditions outside the laboratory. This is because the larvae were reared in laboratory conditions with controlled diet for five generation. It could be that the observed gut-microbial differences are condition specific and may not represent the actual gut-microbial composition.

3.It is not clear if DNA was extracted from the midgut (Line 108) or the entire gut (Line 110). The diversity of the gut-microbiota varies with the different regions of the insect gut. Thus, it is ideal to analyze the gut-microbial populations of the foregut, midgut and hindgut regions seperately for a complete understandings of the region-specific gut-microbial diversity.

4. Please mention in the manuscipt if the data is a culmination of three independent biological experiments.

Validity of the findings

The research findings are novel and are likely to have an impact in this field of research. Conclusions are well stated and linked to original research question..

Additional comments

Minor comments:
1. Line 175 – Anopheles? Please correct.
2. Figure 1: OTU-Full form
3. Title: The authors may consider writing the S. litura in full form.

Reviewer 2 ·

Basic reporting

The manuscript is very well english written. The references used in the text are enough to introduce the theme and discuss the results. All the figures attached are on good quality and the structure of the manuscript is good
I have some comments about the text:

- You should write S. frugiperda and S. litura in italic. Review all names in the text.
- Line 37: remove "and so on."
- Line 51: There is a mistake in the typing
- Lines 114-116: How much the DNA concentration that you used to sequence by Illumina?
- Change "flora" to "microbiome"
- Line 222: add the citation to this sentence
- Lines 235-236: Please, you should rewrite this sentence. Peptostreptococcaceae and Clostridiales are not genera.
- Line 239: "Drosophila" is italic
- LIne 242: Please, how do you know this information about gene expression in your study? I understand that you did only DNA extraction, not RNA extration.

Experimental design

no comments

Validity of the findings

no comment

Annotated reviews are not available for download in order to protect the identity of reviewers who chose to remain anonymous.

---

## Round 0.2 · accepted · Accept

The authors addressed all the concerns previously raised by the reviewers. Consequently, it is now suitable for publication.

Reviewer 1 ·

Basic reporting

The authors have addressed the points raised during the first revision. I have no further comments.

Experimental design

The authors have addressed the points raised during the first revision. I have no further comments.

Validity of the findings

The authors have addressed the points raised during the first revision. I have no further comments.

Reviewer 2 ·

Basic reporting

The authors of the manuscript revised all text and accepted the comments from reviewers.

Experimental design

The authors of the manuscript revised all text and accepted the comments from reviewers.

Validity of the findings

The authors of the manuscript revised all text and accepted the comments from reviewers.